# The large-scale blast score ratio (LS-BSR) pipeline: a method to rapidly compare genetic content between bacterial genomes

Jason W. Sahl[1,2,4], J. Gregory Caporaso[2], David A. Rasko[3] and Paul Keim[1,2,4]

[1] Division of Pathogen Genomics, Translational Genomics Research Institute, Flagstaff, AZ, USA
[2] Department of Biological Sciences, Northern Arizona University, Flagstaff, AZ, USA
[3] Department of Microbiology and Immunology, Institute for Genome Sciences, University of Maryland School of Medicine, Baltimore, MD, USA
[4] Center for Microbial Genetics and Genomics, Northern Arizona University, Flagstaff, AZ, USA

Corresponding author
Jason W. Sahl, jsahl@tgen.org

## ABSTRACT

**Background.** As whole genome sequence data from bacterial isolates becomes cheaper to generate, computational methods are needed to correlate sequence data with biological observations. Here we present the large-scale BLAST score ratio (LS-BSR) pipeline, which rapidly compares the genetic content of hundreds to thousands of bacterial genomes, and returns a matrix that describes the relatedness of all coding sequences (CDSs) in all genomes surveyed. This matrix can be easily parsed in order to identify genetic relationships between bacterial genomes. Although pipelines have been published that group peptides by sequence similarity, no other software performs the rapid, large-scale, full-genome comparative analyses carried out by LS-BSR.

**Results.** To demonstrate the utility of the method, the LS-BSR pipeline was tested on 96 *Escherichia coli* and *Shigella* genomes; the pipeline ran in 163 min using 16 processors, which is a greater than 7-fold speedup compared to using a single processor. The BSR values for each CDS, which indicate a relative level of relatedness, were then mapped to each genome on an independent core genome single nucleotide polymorphism (SNP) based phylogeny. Comparisons were then used to identify clade specific CDS markers and validate the LS-BSR pipeline based on molecular markers that delineate between classical *E. coli* pathogenic variant (pathovar) designations. Scalability tests demonstrated that the LS-BSR pipeline can process 1,000 *E. coli* genomes in 27–57 h, depending upon the alignment method, using 16 processors.

**Conclusions.** LS-BSR is an open-source, parallel implementation of the BSR algorithm, enabling rapid comparison of the genetic content of large numbers of genomes. The results of the pipeline can be used to identify specific markers between user-defined phylogenetic groups, and to identify the loss and/or acquisition of genetic information between bacterial isolates. Taxa-specific genetic markers can then be translated into clinical diagnostics, or can be used to identify broadly conserved putative therapeutic candidates.

## INTRODUCTION

Whole genome sequence (WGS) data has changed our view of bacterial relatedness and evolution. Computational analyses available for WGS data include, but are not limited to, single nucleotide polymorphism (SNP) discovery (*DePristo et al., 2011*), core genome phylogenetics (*Sahl et al., 2011*), and gene based comparative methods (*Hazen et al., 2013*; *Sahl et al., 2013*). In 2005, a BLAST score ratio (BSR) method was introduced in order to compare peptide identity from a limited number of bacterial genomes (*Rasko, Myers & Ravel, 2005*). However, the "all vs. all" implementation of this method scales poorly with a larger number of sequenced genomes.

Here we present the Large Scale BSR method (LS-BSR) that can rapidly compare gene content of a large number of bacterial genomes. Comparable methods have been published in order to group genes into gene families, including OrthoMCL (*Li, Stoeckert & Roos, 2003*), TribeMCL (*Enright, Van Dongen & Ouzounis, 2002*), and GETHOGs (*Altenhoff et al., 2013*). Although grouping peptides into gene families is not the primary focus of LS-BSR, the output can be parsed to identify the pan-genome (*Tettelin et al., 2008*) structure of a species; scripts are included with LS-BSR that classify coding sequences (CDSs) into pan-genome categories based on user-defined identity thresholds.

Pipelines have also been established to perform comprehensive pan-genome analyses, including the pan-genome analysis pipeline (PGAP) (*Zhao et al., 2012*), which requires specific gene annotation from GenBank and complicates the analysis of large numbers of novel genomes. PGAP also doesn't allow for the screen of specific genes of interest against query genomes in order to identify patterns of distribution. GET_HOMOLOGUES (*Contreras-Moreira & Vinuesa, 2013*) is a recently published tool that can be used for pan-genome analyses, including the generation of dendrograms based on the presence/absence of homologous genes; by only using presence/absence based on gene homology, more distantly related gene relatedness cannot be fully investigated. The integrated toolkit for the exploration of microbial pan-genomes (ITEP) toolkit (*Benedict et al., 2014*) was recently published and performs similar functions to LS-BSR, including the identification of gene gain/loss at nodes of a phylogeny. ITEP relies on multiple dependencies and workflows, which are available as a pre-packaged virtual machine. The authors of ITEP report that an analysis of 200 diverse genomes would take ~6 days on a server with 12 processors and scales quadratically with additional genomes.

## MATERIALS AND METHODS

The LS-BSR method can either use a defined set of genes, or can use Prodigal (*Hyatt et al., 2010*) to predict CDSs from a set of query genomes. When using Prodigal, all CDSs are concatenated and then de-replicated using USEARCH (*Edgar, 2010*) at a pairwise identity of 0.9 (identity threshold can be modified by the user). Each unique CDS is then translated with BioPython (www.biopython.org) and aligned against its nucleotide

Peer**J**

sequence with TBLASTN (*Altschul et al., 1997*) to calculate the reference bit score; if BLASTN or BLAT (*Kent, 2002*) is invoked, the nucleotide sequences are aligned. Each query sequence is then aligned against each genome with BLAT, BLASTN, or TBLASTN and the query bit score is tabulated. The BSR value is calculated by dividing the query bit score by the reference bit score, resulting in a BSR value between 0.0 and 1.0 (values slightly higher than 1.0 have been observed due to variable bit score values obtained by TBLASTN). The results of the LS-BSR pipeline include a matrix that contains each unique CDS name and the BSR value in each genome surveyed. CDSs that have more than one significant BSR value in at least one genome are also identified in the output. A separate file is generated for CDSs where one duplicate is significantly different than the other in at least one genome; these regions could represent paralogs and may require further detailed investigation. Once the LS-BSR matrix is generated, the results can easily be visualized as a heatmap or cluster with the Multiple Experiment Viewer (MeV) (*Saeed et al., 2006*) or R (*R Core Team, 2013*); the heatmap represents a visual depiction of the relatedness of all peptides in the pan-genome across all genomes. The Interactive Tree Of Life project (*Bork et al., 2008*) can also be used to generate heatmaps from LS-BSR output and correlate heatmap data with a provided phylogeny. A script is included with LS-BSR (compare_BSR.py) to rapidly compare CDSs between user-defined sub-groups, using a range of BSR thresholds set for CDS presence/absence. Annotation of identified CDSs can then be applied using tools including RAST (*Aziz et al., 2008*) and prokka (http://www.vicbioinformatics.com/software.prokka.shtml). LS-BSR source code, unit tests, and test data can be freely obtained at https://github.com/jasonsahl/LS-BSR under a GNU GPL v3 license.

## RESULTS AND DISCUSSION

### LS-BSR algorithm speed and scalability

To determine the scalability of the LS-BSR method, 1,000 *Escherichia coli* and *Shigella* genomes were downloaded from Genbank (*Benson et al., 2012*); *E. coli* was used as a test case due to the large number of genomes deposited in Genbank. Genomes were sub-sampled at different depths (100 through 1000, sampling every 100) with a python script (https://gist.github.com/jasonsahl/115d22bfa35ac932d452) and processed with LS-BSR using 16 processors. A plot of wall time and the number of genomes processed demonstrates the scalability of the method (Fig. 1A) using three different alignment methods. To demonstrate the parallel nature of the algorithm, 100 *E. coli* genomes were processed with different numbers of processors. The results demonstrate decreased runtime of LS-BSR with an increase in the number of processors used (Fig. 1B).

### Improvements on a previous BSR implementation

The LS-BSR method is an improvement on a previous BSR implementation (http://bsr.igs.umaryland.edu/) in terms of speed and ease of use. The former BSR algorithm (*Rasko, Myers & Ravel, 2005*) requires peptide sequences and genomic coordinates of CDSs to run. LS-BSR only requires genome assemblies in FASTA format, which is the standard

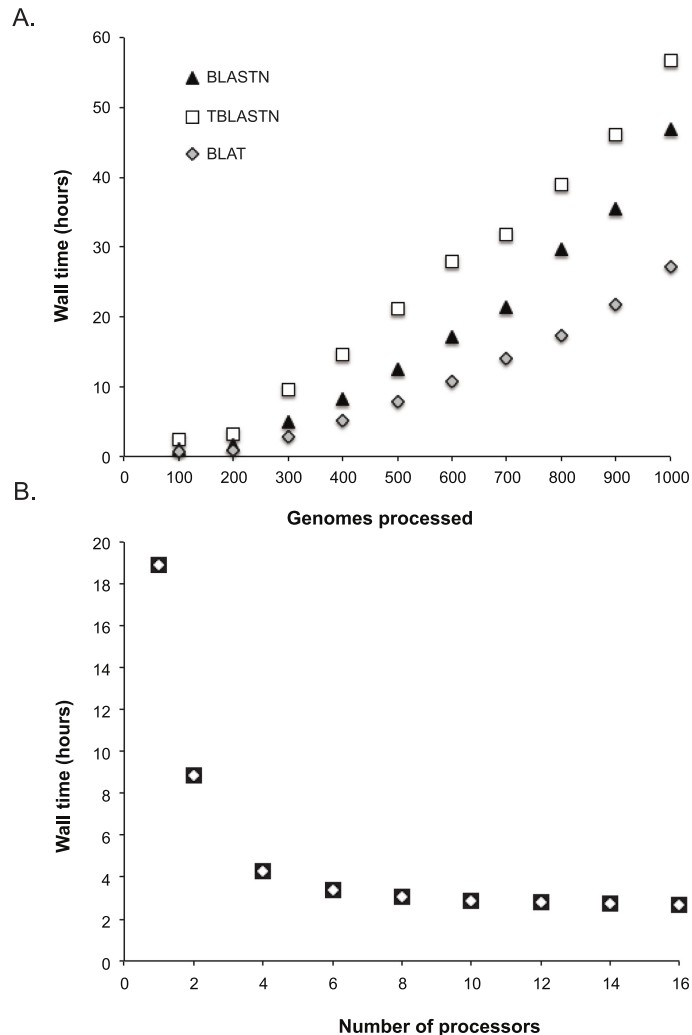

**Figure 1** **Time performance of the LS-BSR pipeline.** (A) 1000 *Escherichia coli* and *Shigella* genomes were randomly sub-sampled and analyzed using default LS-BSR parameters and 16 processors. Wall time was plotted against the number of genomes analyzed. The results demonstrate that the LS-BSR pipeline scales well with increasing numbers of genomes. (B) The same set of 100 *E. coli* genomes was processed with different numbers of processors and the wall time was plotted. The results demonstrate that using additional processors decreases the overall run time of LS-BSR.

output of most genome assemblers. To test the speed differences between methods, 10 *E. coli* genomes (Table S1) were processed with both methods. Using the same number of processors ($n = 2$) on the same server, the original BSR method took $\sim$14 h (wall time) to complete, while the LS-BSR method, using TBLASTN, took $\sim$25 min to complete (wall time). Because the original BSR method is an "all vs. all" comparison and the LS-BSR method is a "one vs. all" comparison, this difference is expected to be more pronounced as the number of genomes analyzed increases.

## Test case: analysis of 96 *E. coli* and *Shigella* genomes

To demonstrate the utility of the LS-BSR pipeline, a set of 96 *E. coli* and *Shigella* genomes were processed (Table S1); these genomes are in various stages of assembly completeness and have been generated with various sequencing technologies from Sanger to Illumina. The BSR matrix was generated with TBLASTN in 2 h 34 min from a set of ∼20,000 unique CDSs using 16 processors. In addition to the LS-BSR analysis, a core genome single nucleotide polymorphism (SNP) phylogeny was inferred on 96 genomes using methods published previously (*Sahl et al., 2011*); the SNP phylogeny with labels is shown in Fig. S1. Briefly, all genomes were aligned with Mugsy (*Angiuoli & Salzberg, 2011*) and the core genome was extracted from the whole genome alignment; the alignment file was then converted into a multiple sequence alignment in FASTA format. Gaps in the alignment were removed with Mothur (*Schloss et al., 2009*) and a phylogeny was inferred on the reduced alignment with FastTree2 (*Price, Dehal & Arkin, 2010*).

The compare_ BSR.py script included with LS-BSR was used to identity CDS markers that are unique to specific phylogenetic clades (Fig. 2). Identified CDSs had a BSR value ⩾0.8 in targeted genomes and a BSR value <0.4 in non-targeted genomes; the gene annotation of all marker CDSs is detailed in Table S2. The conservation and distribution of all clade-specific markers was visualized by correlating the phylogeny with a heatmap of BSR values (Fig. 2). This presentation provides an easy way for the user to highlight features conserved in one or more phylogenomic clades.

*E. coli* and *Shigella* pathogenic variants (pathovars) are delineated by the presence of genetic markers primarily present on mobile genetic elements (*Rasko et al., 2008*). The conservation of these markers was used as a validation of the LS-BSR method. A representative sequence from each pathovar-specific marker (Table S2) was screened against the 96-genome test set and the BSR values (Table S3) were visualized as a heatmap (Fig. 2). The BSR matrix demonstrates that pathovar-specific genes were accurately identified in each targeted genome (Table S3, Fig. 2). For example, the *ipaH3* marker was positively identified in all *Shigella* genomes and the Shiga toxin gene (*stx2a*) was conserved in the clade including O157:H7 *E. coli* (Fig. 2). A sub-set of these 96 *E. coli* genomes is included with LS-BSR as test data to characterize the conservation and distribution of pathovar specific genes.

Finally, the BSR values were used to cluster all 96 genomes with an average linkage algorithm implemented in MeV and the structure of the resulting dendrogram was compared to the core SNP phylogeny. The BSR based clustering method incorporates both the core and accessory genome, while the SNP phylogeny relies on core genomic regions alone. A comparison of the tree structures demonstrates that while *Shigella* genomes share a diverse evolutionary history (Fig. 3A), they all cluster together based on gene presence and conservation (Fig. 3B). This result was also observed using a k-mer frequency method (*Sims & Kim, 2011*), which uses all possible k-mer values to infer a phylogeny and validates the findings of the LS-BSR pipeline. The dendrogram also differed from the core SNP phylogeny in other genomes, which could represent either assembly problems, or more likely the acquisition of accessory genomic regions that are not a product of direct descent.

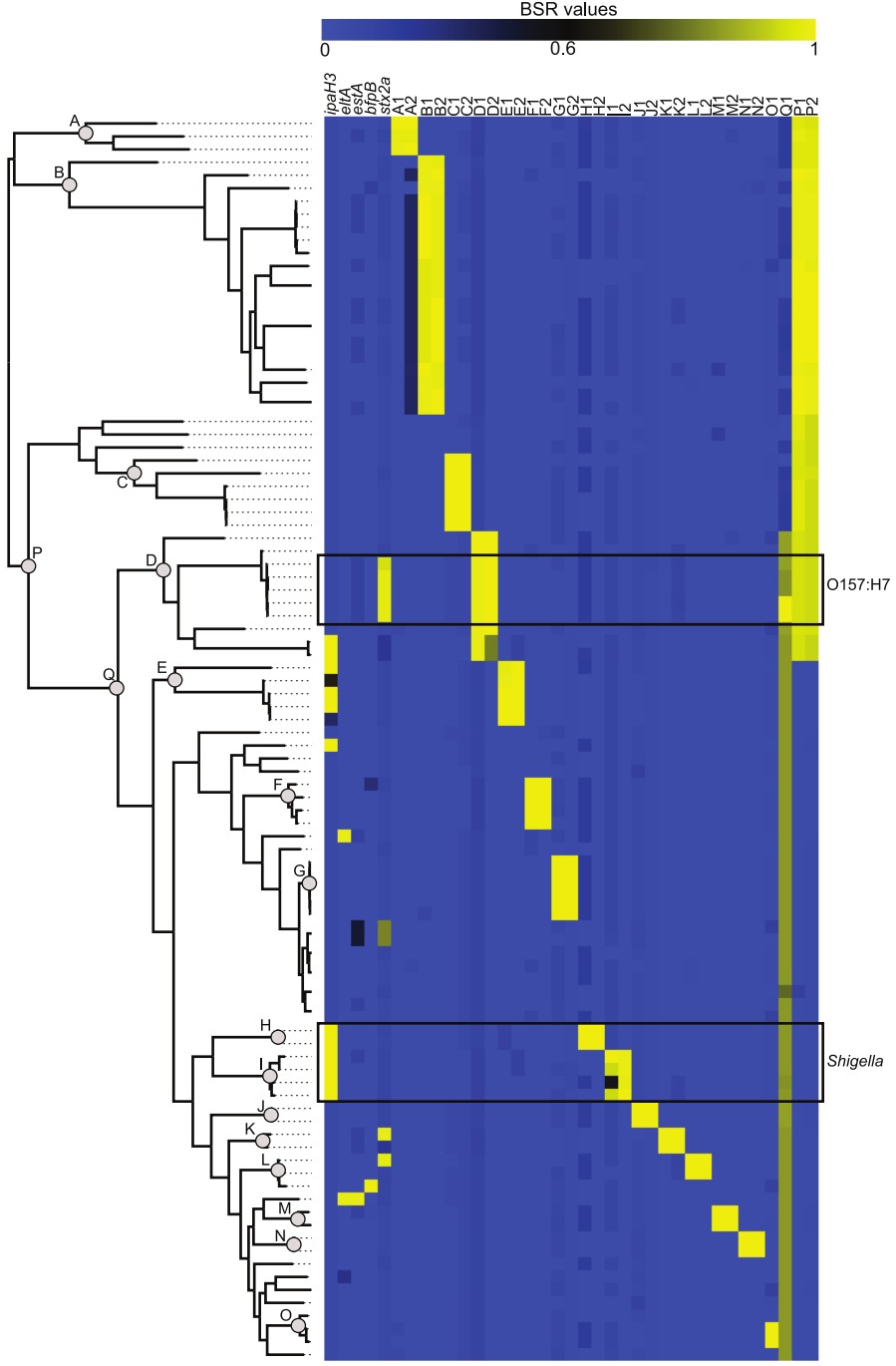

**Figure 2 The distribution of virulence factors and phylogenomic markers associated with a core single nucleotide polymorphism (SNP) phylogeny.** The core SNP phylogeny was inferred from a whole genome alignment produced by Mugsy (*Angiuoli & Salzberg, 2011*). Known virulence genes (Table S2) were screened against 96 *Escherichia coli* and *Shigella* genomes using BLASTN within LS-BSR. Clade specific markers were identified at defined nodes in the phylogeny (A through Q). Gene annotations for these markers are detailed in Table S2.

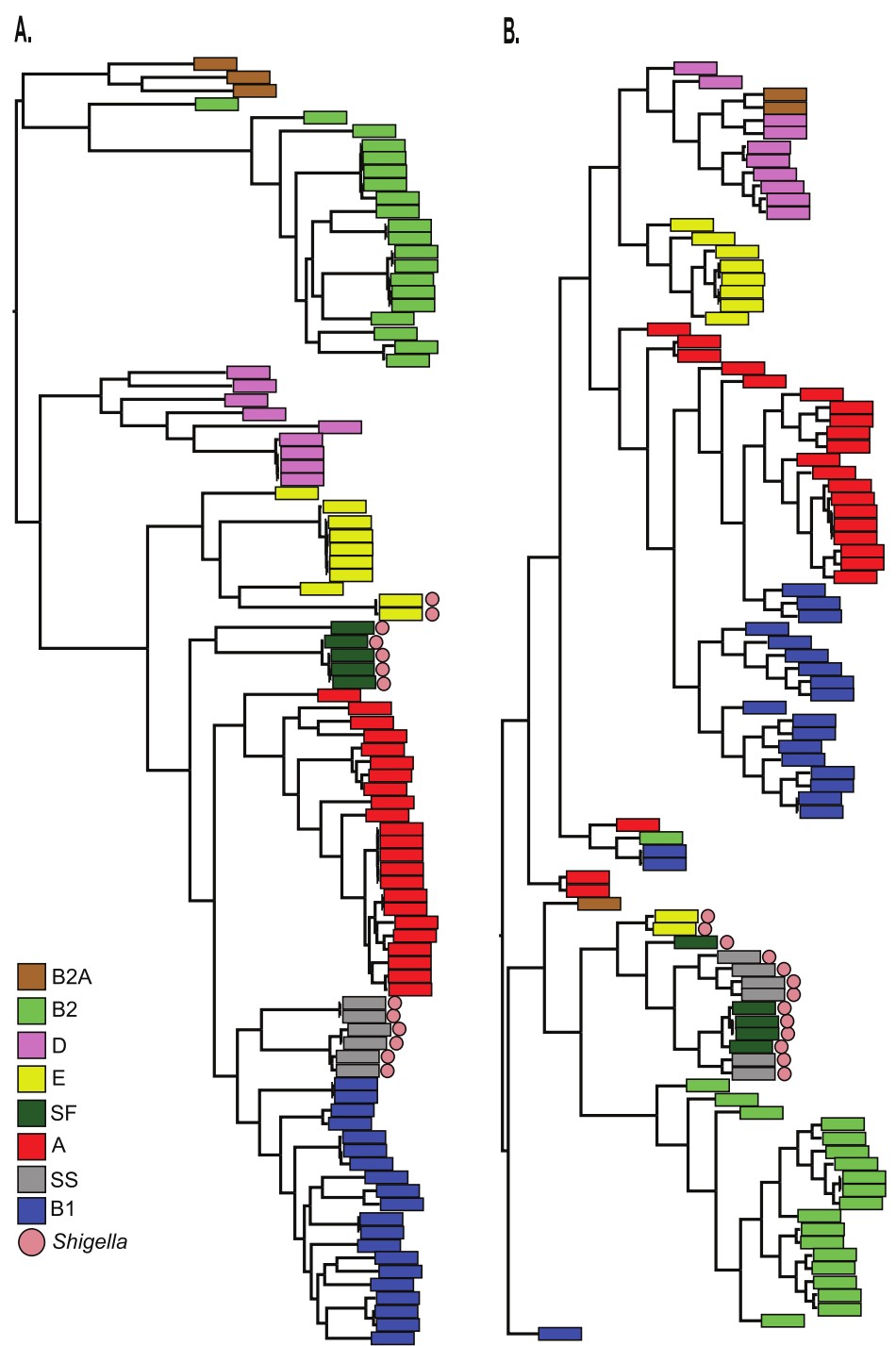

| | |
|---|---|
| ■ | B2A |
| ■ | B2 |
| ■ | D |
| ■ | E |
| ■ | SF |
| ■ | A |
| ■ | SS |
| ■ | B1 |
| ○ | *Shigella* |

**Figure 3 Comparison of LS-BSR cluster with core genome SNP phylogeny.** A comparison of 96 *Escherichia coli/Shigella* genomes between (A) a core single nucleotide polymorphism (SNP) phylogeny or (B) a cluster generated with the Multiple Experiment Viewer (*Saeed et al., 2006*) from BLAST Score Ratio (BSR) values that include the entire pan-genome. Colors applied to each classical *E. coli* phylogroup were applied to the SNP phylogeny and transferred to the BSR cladogram. *Shigella* genomes are marked with a red circle.
**Table 1** Comparison of four pan-genome methods on a test set of 11 *Streptococcus pyogenes* genomes.

|  | LS-BSR | GET_HOMOLOGUES | PGAP | ITEP |
|---|---|---|---|---|
| Clusters orthologs? | Yes | Yes | Yes | Yes |
| Open source? | Yes | Yes | Yes | Yes |
| Pan-genome calculation? | Yes | Yes | Yes | Yes |
| Lineage specific gene identification? | Yes | Yes | Yes | Yes |
| Functional annotation? | No | Yes | Yes | Yes |
| Analyzes user-defined genes? | Yes | No | No | Yes |
| Input files | ".fasta" | GenBank or ".faa" | ".faa", ".fna", ".ppt" | GenBank |
| Supported platforms | linux, OSX | linux/OSX | linux | linux/OSX |
| Core genome size | 1318, 1350, 1426[a] | 1232, 1234[b] | 1332, 1366[c,d] | 1342 |
| Time (2 cores), only runtime | 5 m 59 s, 1 m 53 s, 1 m 17 s[a] | 25 m 14 s | 29 m 59 s,199 m 58 s[c,d] | 24 m 22 s |

**Notes.**
[a] TBLASTN, BLASTN, BLAT.
[b] COG, MCL.
[c] MP, GF.
[d] Taken from publication.

The functionality of LS-BSR was compared to recently released pan-genome software packages including GET_HOMOLOGUES (*Contreras-Moreira & Vinuesa, 2013*), ITEP (*Benedict et al., 2014*), and PGAP (*Zhao et al., 2012*). A set of 11 *Streptococcus pyogenes* genomes was chosen for the comparative analysis, as it was also used as a test set in the PGAP publication; the comparative analysis and results are shown in Table 1. Overall, the size of the core genome was comparable between methods, with LS-BSR (BLASTN) and GET_HOMOLOGUES calculating differing core genome numbers compared to the other methods. However, small differences were expected due to differing thresholds and clustering algorithms. Based on these results, LS-BSR represent a significant improvement in terms of speed and ease of use compared to comparable methods, while having comparable utility.

## Pan-genome analyses

One application in comparative genomics is the analysis of the pan-genome, or the combined genome, of isolates within a species. Post matrix-building scripts are available to visualize the pan-genome of a given dataset. One script (BSR_ to_ PANGP.py) creates a matrix compatible with PanGP (*Zhao et al., 2014*), for visualization of pan-genome statistics. The pan_ genome_ stats.py script provides data that can be used to visualize the conservation of CDSs at different genome depths (Fig. 4A). An additional script randomly subsamples the CDS distribution at all depths and produces data that can be plotted to visualize core genome convergence (Fig. 4B), accumulation of CDSs (Fig. 4C), and the number of unique CDSs for each genome analyzed (Fig. 4D). All analyses were conducted on a set of 100 *E. coli* genomes, with 100 iterations.

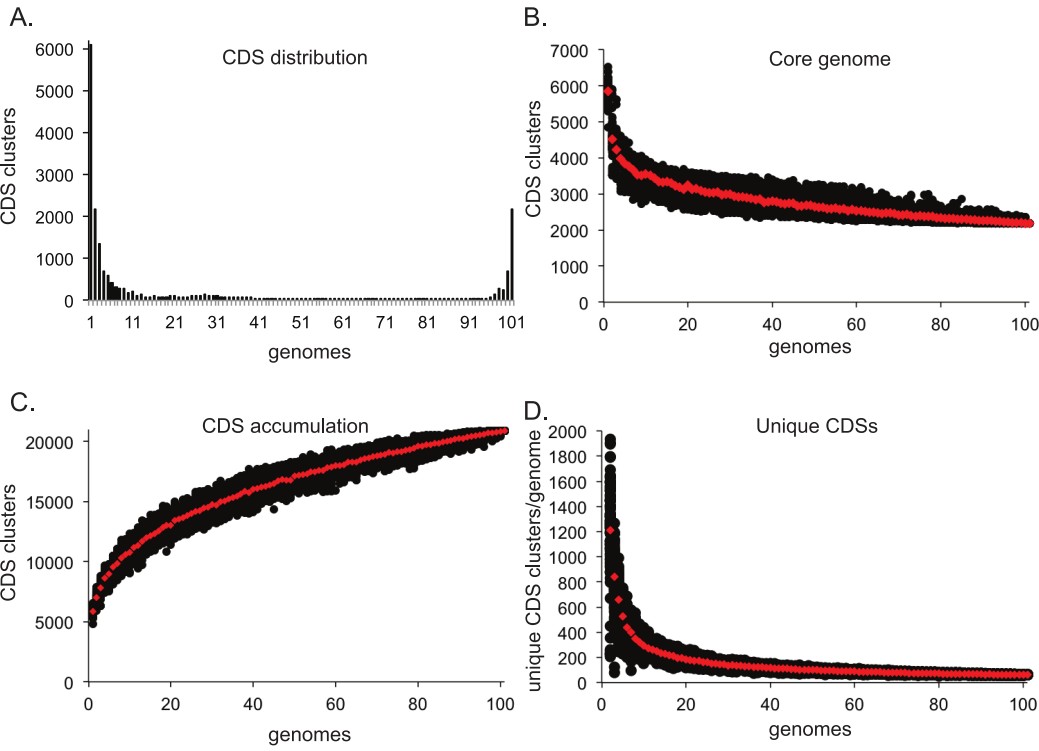

**Figure 4  Pan-genome plots generated from LS-BSR output.** Analyses were conducted on a set of 100 *Escherichia coli* genomes. The distribution of coding region sequences (CDSs) across the set of genomes surveyed is shown in A. A supplemental script can be used to better understand the convergence of the core genome (B), the accumulation of CDSs (C), and the number of unique CDSs for each genome analyzed (D); each analysis was conducted with 100 random sum-samplings and means are depicted with red diamonds.

## CONCLUSIONS

The LS-BSR method can rapidly compare the gene content of a relatively large number of bacterial genomes in either draft or complete form, though with more fragmented assemblies LS-BSR is likely to perform sub-optimally. As sequence read lengths improve, assembly fragmentation should become less problematic due to more contiguous assemblies. LS-BSR can also be used to rapidly screen a collection of genomes for the conservation of known virulence factors or genetic features. By using a range of peptide relatedness, instead of a defined threshold, homologs and paralogs can also be identified for further characterization.

LS-BSR is written in Python with many steps conducted in parallel. This allows the script to scale well from hundreds to thousands of genomes. The LS-BSR method is a major improvement on a previous BSR implementation in terms of speed, ease of use, and utility. As more WGS data from bacterial genomes become available, methods will be required to quickly compare their genetic content and perform pan-genome analyses. LS-BSR is an open-source software package to rapidly perform these comparative genomic workflows.

## ACKNOWLEDGEMENTS

Thanks to Darrin Lemmer for his critical review of the LS-BSR code.

### Funding

This work was funded by the NAU Technology and Research Initiative Fund (TRIF). The funders had no role in study design, data collection and analysis, decision to publish, or preparation of the manuscript.

### Grant Disclosures

The following grant information was disclosed by the authors:
NAU Technology and Research Initiative Fund (TRIF).

### Competing Interests

Jason W. Sahl is employed by The Translational Genomics Research Institute, and Paul S. Keim is the Director of The Translational Genomics Research Institute.

### Author Contributions

- Jason W. Sahl conceived and designed the experiments, performed the experiments, analyzed the data, contributed reagents/materials/analysis tools, wrote the paper, prepared figures and/or tables, reviewed drafts of the paper.
- J. Gregory Caporaso analyzed the data.
- David A. Rasko conceived and designed the experiments, contributed reagents/materials/analysis tools.
- Paul Keim conceived and designed the experiments.

### Supplemental Information

Supplemental information for this article can be found online at http://dx.doi.org/10.7717/peerj.332.

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
