# Peer review of "The large-scale blast score ratio (LS-BSR) pipeline: a method to rapidly compare genetic content between bacterial genomes"

_PeerJ, doi:10.7717/peerj.332_

## Round 0.1 · original submission · Major Revisions

Please address every issue mentioned by the two reviewers.

·

Basic reporting

No comments

Experimental design

No comments

Validity of the findings

No comments

Additional comments

I was very pleased to receive this well-written manuscript describing a method for pan-genome computation for hundreds to thousands of bacterial genomes. Such software is very much needed in this field. Although the method is technically a pipeline based on existing methods and could be regarded as not particularly novel, the implementation of the software is excellent. The main innovation seems to be the use of UCLUST, and the integration of the BSR method previously described by the authors.

I found the software easy to download, install and get running and generate useful results on my own draft genome datasets. I was pleased to see that the source code was stored in a Github repository.

My only concern is that the pipeline could exceed the built-in memory limitations of the UCLUST 32-bit method which would then require buying a license to the 64-bit version when applied to very large numbers of genomes. It would be have been good if an open-source implementation of a clustering method could be specified as an alternative.

I would have also liked to have seen support for faster alternatives to BLAST which could speed the pipeline up hugely whilst providing similar levels of sensitivity (e.g. Rapsearch2, LAST)

If I was to make one request it would be that some example R scripts to generate cluster visualisations from the LS-BSR output could be supplied. Although this is fairly trivial to do for users experienced with R, it would be useful for beginners wishing to generate heatmaps.

One other request: the authors are quite specific about which versions of the dependent software works with LS-BSR. It would be good to specify download links to these particular versions in the documentation, and perhaps document why - particularly in the case of BLAST - later versions of the software do not work.

Reviewer 2 ·

Basic reporting

No Comments

Experimental design

No Comments

Validity of the findings

No Comments

Additional comments

The authors present a nice pipeline for users to compare bacterial genetic content, and it has obvious advantages in speed and calculation. While I have some minor questions about the work.

Comment 1: The authors did not make a comparison on accuracy between LS-BSR and BSR, comparison on speed, accuracy and other feature between LS-BSR and other similar algorithms or methods. I think these are very necessary.

Comment 2: In Figure 1B, wall time was plotted against number of processors. There is slightly difference when using more than 8 processors. I’m wondering if the best choice of processors used is relevant with genome numbers processed. I would like that to be discussed in the manuscript (or set recommended value in the software).

Comment 3: Author described a LS-BSR pipeline along with a set of python scripts (compare_BSR_matrices.py et.al) in the manuscript. However, I cannot find a user manual in the github page, but only the installation guide and list of tools. A friendly user manual or tutorial is important for new users.

---

## Round 0.2 · accepted · Accept

Congratulations on your great work.